

# Brief communication: Strengthening coherence between climate change adaptation and disaster risk reduction through policies, methods and practices in Europe

Jaroslav Mysiak[1], Sergio Castellari[2,8], Blaz Kurnik[2], Rob Swart[3], Patrick Pringle[4], Reimund Schwarze[5],
Henk Wolters[6], Ad Jeuken[6] and Paul van der Linden[7]

[1] Euro-Mediterranean Centre on Climate Change and Università Ca' Foscari, Venezia Marghera, 30175, Italy
[2] European Environment Agency, Copenhagen, 1050, Denmark
[3] Wageningen Environmental Research, Wageningen, 6708PB, The Netherlands
[4] Climate Analytics GmbH, Berlin, 10969 Germany and the Secretariat of the Pacific Regional Environment Programme
(SPREP), Apia, Samoa.
[5] Helmholtz-Zentrum für Umweltforschung – UFZ, Leipzig, 04318, Germany
[6] Deltares, Delft, 2600 MH, The Netherlands
[7] Met Office, Exeter, Devon, EX1 3PB, United Kingdom
[8] Istituto Nazionale di Geofisica e Vulcanologia, Bologna, 40100, Italy

*Correspondence to*: Jaroslav Mysiak (jaroslav.mysiak@cmcc.it)

**Abstract.** Reducing natural hazard risks and adapting to climate change are ever more important policy goals. Sound climate risk management will lessen the impacts of disaster risks and contribute to boosting resilience. Climate change adaptation and disaster risk reduction have to some extent been mainstreamed into international and national policies but it is important to ensure that the resulting efforts are consistent and mutually supportive. The EEA report »Climate change adaptation and
disaster risk reduction in Europe: enhancing coherence of the knowledge base, policies and practices« identifies several ways how the coherence between CCA and DRR can be promoted.

## 1 Introduction

Reducing natural hazard risks and adapting to climate change are ever more important policy goals. The already high economic, social and environmental tolls of extreme weather and climate-related events (hereafter, climate extremes) are
expected to increase sizeably as a result of human-induced climate change (EEA, 2016).

Total reported economic losses from climate extremes in the EEA member countries over the 1980–2016 period come to more than EUR 436 billion (EEA, 2017b). These should be understood as minimal conservative estimates of the actual economic impacts, since many intangible and non-monetary impacts are difficult to quantify. Climate variability and change have noticeable effects on human health, the spread of climate-sensitive diseases, environmental quality and social well-
being, and these impacts are either not or only partially accounted for. In addition, climate extremes can affect and shape ecosystems and thus have an impact on the services they provide (e.g. water retention, food production, cooling, energy production, recreation and carbon sequestration). In some cases the loss of such services can increase the probability of



further hazards. For example, forest fires exacerbated by drought can lead to slope destabilisation and increase the risk of landslides during extreme rainfall events.

Under future climate change, nearly all climate extremes are projected to increase in severity, duration and/or extent, and some also in frequency (EEA, 2016). Heat waves are projected to become more intense and to persist longer in all regions in

Europe (Russo et al., 2014). However, projected changes in frequency and intensity of extreme precipitation show regional differences with the largest increases in central and eastern Europe in winter months (Jacob et al., 2014). Similarly, changes in river floods show strong regional differences with the greatest increases for the British Isles, north-west and south-east France, northern Italy, parts of Spain, the Balkans and the Carpathians (Alfieri et al., 2015; Russo et al., 2014). Along with climate change, growing population and wealth, developments in hazard-prone areas, and the deteriorated status of natural

ecosystems drive the impacts upwards by increasing exposure and vulnerability.

Sound climate risk management will lessen the impacts of disaster risks and contribute to boosting resilience, while harnessing the best available knowledge. Climate change adaptation (CCA) and disaster risk reduction (DRR) have to some extent already been mainstreamed into international, European and national policies, but it is important to ensure that the resulting efforts are internally consistent and mutually supportive (EEA, 2017a). Policy coherence underlies the evaluation

of fitness-for-purpose of European policies (EC, 2017a) and is a cornerstone of the European Consensus on Development and Humanitarian Aid. But there is a remarkable vagueness about how policy coherence between CCA and DRR can be actively promoted. According to a recent report of the European Environment Agency (EEA), evidence of good practices does exist, but public policies, methods and practices often lack coherence, and opportunities are not fully exploited to build up resilience by a better integration of CCA and DRR.

The EEA report »Climate change adaptation and disaster risk reduction in Europe: enhancing coherence of the knowledge base, policies and practices« (EEA, 2017a) tries to fill the above gap. The Report builds upon an extensive desk review, consultations with the European Environment Information and Observation Network (EIONET), and an international expert workshop held in April 2016. Successively, it was comprehensively reviewed by a number of European experts and EEA member country representatives. The Report is a result of joint efforts of the EEA, the European Topic Centre on Climate

Change Impacts, Vulnerability and Adaptation (ETC/CCA), and the European Topic Centre on Inland, Coastal and Marine waters (ETC/ICM), and various services of the European Commission, such as the Joint Research Centre, DG Climate Action and DG Humanitarian Aid and Civil Protection. The release of the Report was aligned in a timely way with complementary reports of the Disaster Risk Management Knowledge Centre and the UN Office for Disaster Risk reduction (Poljanšek et al., 2017; UNISDR, 2017). It sets out to inform an ongoing review of the EU Climate Adaptation Strategy (EC,

2013a) and the implementation of the Sendai Framework for DRR 2015-2030 in the EU (EC, 2016a).

## 2. Defining good examples of policies, methods and practices

*Policies driving CCA and DRR in Europe*



Closer coherence between CCA and DRR is built into and endorsed by the Sendai Framework for Disaster Risk Reduction 2015-2030 (SFDRR; UN, 2015a) and the Paris Agreement on Climate Change (UNFCCC, 2015). Both are related and contribute to the 2030 Agenda for Sustainable Development (UN, 2015b). The SFDRR advocates multi-hazard, inclusive, science-based and risk-informed decision-making, and lays down priorities for action and policy targets. Progress in

achieving these targets is monitored and assessed by means of 38 indicators, some of which are also used to report on the Sustainable Development Goals. The Sendai Framework Monitor was launched in March 2018 to facilitate the reporting. The Paris Agreement specifies, among other things, a global adaptation goal focussed on the ability to adapt to the adverse impacts of climate change and on climate resilience, both among the essential prerequisites of sustainable development.

The EU Action Plan on SFDRR (EC, 2016a) acknowledged that a coherent realisation of the objectives laid down in the

Framework could not only boost resilience, but also spur innovation and growth in the context of sustainable development. The EU's Civil Protection Mechanism (EC, 2013b) emphasizes multi-hazard (including climate) risk assessments and (short to long-term) prevention as bases for effective disaster preparedness and response. In 2017 the Commission proposed a reform of the Mechanism (rescEU, EC 2017b) which accentuates coherence between climate change adaptation, disaster prevention and disaster response. The proposal empowers the Commission to compel, monitor and report on the

implementation of prevention and preparedness plans (EC, 2017b).

The EU Strategy on Adaptation to Climate Change (EC, 2013a) has fostered development of national adaptation strategies and national adaptation plans and boosted knowledge sharing and mainstreaming of climate adaptation in other policy areas. These policy domains include environment and critical infrastructure protection, agriculture, food and nutrition security, integrated coastal management, Cohesion Policy and EU Structural and Investment Funds. Building upon the achievements

of the Europe 2020 strategy, the EC developed a long-term vision beyond 2020 and invited the EU member states to develop national frameworks for the achievement of the SDGs (EC, 2016b).

*Common methods and concerns*

Assessment of climate-related hazards and risks is an area that has long stimulated the building of common grounds between

CCA and DRR. Climate-related hazards are outcomes of multiple stochastic processes. On a temporal scale, the probability distributions span years, decades and centuries. In some cases, even lower probabilities are still relevant for today's decision making. These stochastic processes are often not stationary but respond to environmental changes, including climate change. This is why CCA and DRR communities have sought and succeeded in reconciling key terms and definitions (Jurgilevich et al., 2017). The levels of vulnerability, which include sensitivity or susceptibility to harm and lack of capacity to cope

and adapt, are changing as our societies are transformed in terms of demography, wealth, cohesion and use of technology. Notwithstanding the importance of the quality-assured, systematically collected and thorough records on impacts of natural hazards, the loss data systems in Europe are fragmented and inconsistent. Empirical and evidence-based risk analysis and assessment are a vital part of CCA and DRR efforts.

*Characteristics identified in good practice examples*





Innovative examples of integration in CCA and DRR policy and practice exist, but are not always explicit, e.g. in the flood risk prevention sector, where often climate projections are already considered in designing strategies and plans. In most areas, however, they are not, and the full potential of a better integration of CCA and DRR has yet to be exploited. This better integration will need closer and more effective vertical and horizontal, cross-border and transnational coordination and

cooperation.

The desk review and consultations have revealed a lack of unambiguous criteria defining 'good' practice examples for fostering the coherence between CCA and DRR actions. Cases were identified which were characterised by a higher level of coherence, as well as examples that hold greater potential for transferable lessons learned. Notwithstanding our extensive network search, we did not find many cases in which enhanced coherence was a planned outcome, with added value for both

policy areas; where uncertainty was given due regard from the long-term perspective; and where the reviewed practices were well embedded within the risk management and climate adaptation planning cycles.

Good examples of governance exhibited a robust legislative mandate, well-defined organisational and institutional tiers, and clearly assigned roles and responsibilities. In terms of risk and adaptation financing, good practices include proper budgetary endowments and sound use of financial or economic instruments/incentives. From a policy perspective, the proposed

measures should not only be efficient and effective, but also compatible with and complementary to measures implemented for other similar purposes. On a more practical level, good practice examples imply use of combined knowledge and data on short and long-term hazards, exposure, vulnerability and performance of past climate risk reduction efforts, including the underlying assumptions and uncertainties.

Six good examples were eventually chosen for the report and include (1) development of a long-term planning vision in the

Netherlands; (2) insurance and risk financing based on public–private partnerships in Spain, France and the United Kingdom; (3) local risk governance in Switzerland; (4) national risk assessments serving both CCA and DRR purposes; (5) city networking for improved urban resilience; and (6) financing nature-based solutions for CCA and DRR (Table 1).

| Good practice examples | and their features |
|---|---|
| Multi-level and long-term governance (e.g. Delta programme in Netherlands) | • Multi-actor partnership for designing climate-proof water management across otherwise separate policy domains,<br>• Multi-layer safety policies and measures in which an optimal mix is proposed between prevention, sustainable spatial planning and crisis management,<br>• Adaptive Delta Management employing short-term interventions within long-term planning perspectives, while taking into account uncertain impacts of climate change through a range of scenarios. |
| Public-private partnerships (PPPs) for hazard risk transfer (e.g. insurance and | • Examples of longstanding insurance-related PPPs include the Spanish Consorcio de Compensación de Seguros (CCS), the French Catastrophes Naturelles (CatNat) and the Flood Reinsurance Scheme (Flood Re) in the UK, |





| | |
|---|---|
| reinsurance schemes in Spain, France, UK) | • Vehicles of joint bearing of responsibilities and efficient risk sharing enabling insurability and financial backing also for low-probability/high-impact risks, <br> • Incentivising risk prevention, helping to improve risk understanding and knowledge, and stimulating active engagement and investment. |
| Multi-level risk governance (e.g. Switzerland) | • Decentralised system with cantons and municipalities investing operational responsibility for DRR and civil protection, and federal authorities engaged in strategic planning, financial and technical support, and overall control, <br> • Formal arrangements secure cooperation between these actors, horizontally and vertically, and between federal organisations, the private sector and academic organisations. |
| National risk assessments (NRAs) | • NRAs are instrumental for identifying, assessing and prioritising security threats, including those arising from climate variability and change, <br> • Experiences of some countries, such as France, the Netherlands and the United Kingdom, show that climate vulnerability and risk assessments need to build on strong institutional frameworks, clearly assigned responsibilities and authority, and close stakeholder engagement. |
| City networks | • Covenant of Mayors for Climate and Energy, C40 Cities, UNISDR Making Cities Resilient campaign and Rockefeller 100 Resilient Cities, and others, <br> • Collaboration in absence of hierarchical authority, building upon information and communication, project funding, capacity building, good practice benchmarking and certification. |
| Financing nature-based solutions (NBS) (e.g. European Investment Bank) | • Ecosystems may mitigate natural hazard risks, by mediation of flows and nuisances; or through maintenance of physical, chemical, biological conditions in the face of pressures, <br> • European Investment Bank's Natural Capital Finance Facility (NCFF) is a new finance instrument which aims specifically at financing projects which apply nature-based solutions to adaptation measures, <br> • NCFF sets out to generate a revenue stream or achieve cost savings in order to pay back the investment; the instrument typically includes an equity-type component to reduce risk, and a technical assistance component. |

**Table 1: Selected examples of good practice that fosters coherence between DRR and CCA**





**3. Opportunities to enhance coherence between climate change adaptation and disaster risk reduction in policy and practice**

Notwithstanding the interconnected goals and multiple synergies CCA and DRR are two different areas of policy and practice, each characterised by its own institutional and legal frameworks, which differ across countries as a result of multiple factors. The coherence between them can be promoted through sharing knowledge, closer collaboration between existing science and policy platforms and networks, and better use of financial resources. Below we summarize the key recommendations in more detail:

**Resilience to climate variability and change provides common ground for CCA and DRR, upon which more coherent actions can be built**. Building the culture and practice of resilience (NRC, 2012) goes beyond reducing the consequences of foreseeable events and builds resilience into systems to recover and adapt when adverse events occur. Climate change actions - both mitigation and adaptation - contribute to closing the 'resilience gap' by helping to avoid unmanageable changes and managing them when they become unavoidable (Bierbaum et al., 2007). DRR contributes to the same end by improving the capacity to cope with the tail ends of climate variability. For DRR, taking into account long-term climate change will enhance preventive responses to risks, while for CCA and the emerging climate services, considering DRR requires a stronger focus on extremes and a better connection between shorter and longer term projections.

**Improved risk assessment methods and mutually beneficial approaches present opportunities for enhancing coherence between the two policy areas**. National risk assessments and national adaptation strategies have been completed by most of the EEA member countries, sometimes in a coordinated manner. Hazard mapping and risk assessment are areas where integration of CCA and DRR is more advanced and recognised as a priority. A thorough understanding of risks, including their cascade and spillover effects, is also important for a better understanding of implicit and explicit government liabilities, and designing comprehensive risk financing strategies. At a strategic level, CCA and DRR objectives should be pulled together and reconciled under the umbrella of long-term national sustainable development strategies. But connecting CCA and DRR at this high strategic level is useless if it is not complemented by concrete actions at the national and local level with adequate resources allocated.

**Web-based knowledge portals and multi-stakeholder coordination platforms can be designed to help communicate and share more a consistent and complementary knowledge for CCA and DRR**. To make the multiple strategies responsive and sensitive to the needs of vulnerable communities, social strata and businesses, national and local multi-stakeholder platforms have been established in many countries across Europe, driving horizontal cooperation and partnerships across public and private spheres. In order to be effective, the multi-stakeholders engagement and partnerships are to be complemented by effective means of sharing and reusing information and knowledge, conducive to a common understanding of vulnerabilities, risks and solutions. Well-managed knowledge portals provide a platform for sharing information and thus can increase the understanding of vulnerabilities and risks, as well as mitigation and adaptation measures.



**A well-functioning system of public and private, user-driven climate services that connect short- and long-term climatic changes can help catalyse an economic and societal transformation that reduces risks and improves societal resilience**. Moreover, climate services may unlock Europe's innovation potential, competitiveness and economic growth. Over the past decades, the climate services have grown in numbers, quality and sophistication. The EU has made large

investments in frontline systems enabling modern meteorological services under the Copernicus Earth observation programme. The Copernicus Climate Change Service (C3S) is one out of six services of the Copernicus service component, designed to deliver knowledge to support adaptation and mitigation policies. But the uptake of climate services for policy and decision making is still relatively modest. Improved alignment of demand-led CCA and DRR climate service products requires decision-makers from both communities to have stronger linkages with each other, as well as with the providers of

climate information and knowledge, and intermediate providers of climate services. The DRR community has a long history of making use of hydro-meteorological services, but there are opportunities to better integrate uncertainty associated with future climate variability and change.

    **Nature-based solutions (NBSs) are a prime example of means for simultaneously reducing natural hazard risks and boosting societal resilience that address both CCA and DRR**. Ecosystems can provide means for mitigating natural

hazard risks and boosting societal resilience, locally or regionally. Compared to engineered or built solutions, ecosystem-based approaches can be cost-effective and have co-benefits, thus becoming increasingly valuable in the face of more frequent and/or severe extreme events. They have an economic value in the context of natural disaster risk reduction, even if no price is actually paid for their provision and/or maintenance. Many ecosystem-based initiatives have been developed for DRR and CCA to respond to societal challenges through innovative actions inspired or supported by nature. However, a

more systematic learning about impacts and effectiveness of ecosystem-based approaches is needed, by taking account of local perceptions and knowledge, and sustained political support, monitoring and funding.

    **Connecting available funding and financing options for CCA and DRR at the EU and other levels can identify new opportunities for projects and programmes enhancing resilience**. To mainstream climate change concerns in its broader development strategy, the EU has agreed to spend 20% of its resources under the Multiannual Financial Framework 2014–

2020 on climate change-related action. Under the European Structural and Investment Fund (ESIF), EUR 29 billion have been allocated to the thematic objective 'Climate change adaptation and risk management' (EC, 2016a), but disaster resilience and climate risk management are also promoted under other priorities. Additional funds available for fostering climate adaptation and DRR include Horizon 2020, LIFE and the European Solidarity Fund. Policy instruments that incentivise a more efficient use of natural resources contribute to reducing the impacts of climate change. Economic

incentives and/or disincentives drive individual and business behaviour toward achieving sustainable development objectives, including an efficient use of natural resources and disaster risk reduction. Pricing instruments such as land taxes, tax reliefs or subsidies are commonly applied to correct market failures and decouple environmental pressures from economic growth. Incentive and transparent pricing (e.g. of insurance policies or water) can contribute to reducing the economic effects of extreme events (such as droughts and floods).





**Setting up an interaction and learning mechanism between emerging Monitoring, Reporting and Evaluation (MRE) schemes can improve coherence, quality and relevance for CCA and DRR**. MRE can help learning across cities, regions and countries. CCA and DRR share a number of characteristics that can make MRE challenging, such as long timescales, uncertainty and common baselines. Improving the connectivity and coordination of national-level indicators between DRR, CCA and other policy frameworks such as the SDGs can improve the efficiency of data collection and build up a more complete picture of CCA and DRR progress and priorities. It can also support improved learning regarding the integration of CCA and DRR, and how this can lead to more efficient and effective implementation on the ground.

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
