# Peer review of "Brief communication: Strengthening coherence between climate change adaptation and disaster risk reduction"

_Natural Hazards and Earth System Sciences, 2018_

## Referee Comment (RC1) · L. Booth (Referee) · 2 May 2018

The paper reads well and accurately pinpoints areas for development between CCA and DRR. The tables are especially useful. Some points: For the multi-level risk governance in Switzerland, perhaps see ESPREssO national report for Switzerland (available via referee or elements of it can be read at: http://www.espressoproject.eu/images/deliverables/ESPREssO_D2.2_FINAL.pdf

plus this paper may help touch on trans-boundary element of synergising CCA and

[Figure]

DRR:

Abad J., Booth L., Marx, S. Ettinger S., Gerard F. (2017) "Comparison of national strategies in France, Germany and Switzerland for DRR and cross-border crisis management.", In Proc. 7th International Conference on Building Resilience; Using scientific knowledge to inform policy and practice in disaster risk reduction, ICBR2017, 27 – 29 November 2017, Bangkok, Thailand.

Emphasis should be placed (e.g. in NRA section, Table 1) of close and maintained stakeholder engagement- it is often easy to get sectors to the table, but not so easy to keep them there, over time.

Section 3. Line 21- direct reference to insurance companies should be made here.

Section 3. Line 24- Maybe emphasise the concrete actions at local level in particular-shows innovation, often under severe financial (or other) restrictions. Perhaps explore examples of poor practice which have generated good/ better practice over time?

Page 7, line 16-17- possible supporting reference for softer solutions with an "in kind" value to CCA made in: Booth & Patt (2018): "The Push for Proactive Climate Adaptation in Europe". Current History Magazine, US p. 108-113.

Page 7, line 28, after Horizon2020, is it appropriate to refer to the ESPREssO Project, one of whose core challenges reflects the theme of this paper and can benefit from its findings?

---

## Referee Comment (RC2) · Anonymous Referee #2 · 29 May 2018

The authors present a brief communication related to the "coherence between CC adaptation and DRR through policies, methods and practices in Europe" based on the recently published EEA report (European Environment Agency, 2017).

While the overall aim of this brief communication is not doubted, it remains a bit open to me (1) what the aim of this manuscript should be apart from advertising the original EEA report, and (2) to what extent the European research landscape is comprehensively mirrored by the authors. As such, most of the cited references are EC and EEA reports and policy briefs, in contrast, sources other than these are widely neglected

(such as e.g., OECD). As such, a brief communication in NHESS could take a wider or more focused point of view than the present one; at the moment the manuscript reads more in the way of an extended abstract of the original report. Consequently, lots of the statements made are not underpinned by references, and if they repeatedly refer to EEA reports again.

The title should in my opinion focus a bit more on the content, which is not "Europe" but the Netherlands, Spain, France, the UK and Switzerland (why this selection?). As such, the manuscript shows good examples (or best practice) of CC adaptation and DRR strategies in European countries, but the focus on some countries means in turn that other examples and good practices are missing. Moreover, the examples shown in Table 1 are also a result of the respective legal regulations in the countries reported. To give an example, the multi-level risk governance reported from Switzerland is a result of the constitution of the Swiss Federation, and not as such the result of specific arrangements in DRR or CC adaptation, and cannot be similarly put to other governmental contexts such as the French one, where decisions are made by definition in a more centralized manner.

In the Introduction section the authors refer to initiatives other than those of EEA, such as those from EC JRC, ETC/CCA, DG Climate Action etc. On the international level, the UN Office for DRR and the Sendai Framework are particularly mentioned. Here, it would be good to also include some challenges behind such European and international actions, as recently shown by Wymann von Dach (2017) or Zimmermann and Keiler (2015) with respect to CC adaptation and DRR for European mountain regions, or OECD (2017). Similarly, it would be nice to see some definitions here (how do the authors of the EEA study define resilience, vulnerability, and risk?). Moreover, some of the facts given are more platitudes than proven scientific facts – at least in the way they are presented here. To give another example, the authors state that "Along with climate change, growing population and wealth, developments in hazard-prone areas, and the deteriorated status of natural ecosystems [...] increasing exposure and vulnerability" can be observed. This is a tricky statement, and has recently been debated (by European scholars) in the scientific literature. The manuscript would surely gain in content and rationale power if we could see some examples here – there are lots of recent works from the UK, the Netherlands or Switzerland and Austria around (such as, for example, Jongman et al. (2014) or Fuchs et al. (2015), both even published in the target journal). This would additionally help to close the still existing gap between policy advice and sciences without narrowing the aims of the original report.

Similarly, section 2 seems to be a bit too narrowly focused on the EC and EEA contexts (page 3, lines 9-21). Consequently, the text is missing some concrete details or particular arguments; sentences such as the "levels of vulnerability, which include sensitivity or susceptibility to harm and lack of capacity to cope and adapt, are changing as our societies are transformed in terms of demography, wealth, cohesion and use of technology. Notwithstanding the importance of the quality-assured, systematically collected and thorough records on impacts of natural hazards, the loss data systems in Europe are fragmented and inconsistent. Empirical and evidence-based risk analysis and assessment are a vital part of CCA and DRR efforts" are a good example, also here there have been made (not only recently) some efforts from the scientific side, spanning from publications such as those of Barredo (2007, 2009, 2010) or Paprotny et al. (2018) on the European scale to national reports from different European countries, all of which are worth being noted. Alternatively, the authors should clearly state at the very beginning that the focus of this manuscript is a selection of European countries and initiatives, with an emphasis on EEA activities.

Finally, section 3 reads like a document of political decision makers with (from a scientific perspective) lots of "common sense" statements, which is of course ok in a framework like the one of the original study, but it would not be amiss to be more specific in the "brief communication" to be published in a scientific journal such as NHESS.

The topic itself is definitely worth being published in NHESS, and I would like to encourage the authors to undertake some revisions in the directions outlined above. As

a result, I kindly would like to suggest to revise this piece of work in order to better show the relation to existing scientific works recently published (also in NHESS), and consequently to lessen the focus on EC and EEA initiatives (this can be gathered easily by the interested reader through the original version of the EEA report (European Environment Agency, 2017)).

References mentioned

Barredo, J.: Major flood disasters in Europe: 1950-2005, Natural Hazards, 42, 125-148, 2007.

Barredo, J.: Normalised flood losses in Europe: 1970-2006, Natural Hazards and Earth System Sciences, 9, 91-104, 2009.

Barredo, J.: No upward trend in normalised windstorm losses in Europe: 1970–2008, Natural Hazards and Earth System Sciences, 10, 97-104, 2010.

European Environment Agency: Climate change adaptation and disaster risk reduction in Europe, EEA Report 15/2017, edited by European Environment Agency, Copenhagen, 172 pp., 2017.

Fuchs, S., Keiler, M., and Zischg, A.: A spatiotemporal multi-hazard exposure assessment based on property data, Natural Hazards and Earth System Sciences, 15, 2127-2142, 2015.

Jongman, B., Koks, E. E., Husby, T. G., and Ward, P. J.: Increasing flood exposure in the Netherlands: implications for risk financing, Natural Hazards and Earth System Sciences, 14, 1245-1255, 2014.

OECD: Boosting disaster prevention through innovative risk governance: Insights from Austria, France and Switzerland, OECD Publishing, Paris, 250 pp., 2017.

Paprotny, D., Morales-Nápoles, O., and Jonkman, S. N.: HANZE: a pan-European database of exposure to natural hazards and damaging historical floods since 1870,

Earth System Science Data, 10, 565-581, 2018.

Wymann von Dach, S., Bachmann, F., Alcántara-Ayala, I., Fuchs, S., Keiler, M., Mishra, A., and Sötz, E.: Safer lives and livelihoods in mountains: Making the Sendai Framework for Disaster Risk Reduction work for sustainable mountain development, Centre for Development and Environment (CDE), University of Bern, with Bern Open Publishing (BOP), Bern, 78 pp., 2017.

Zimmermann, M., and Keiler, M.: International frameworks for disaster risk reduction: Useful guidance for sustainable mountain development?, Mountain Research and Development, 35, 195-202, 2015.

---

## Author Comment (AC1) · 16 Aug 2018

Dear Dr. Laura Booth, thank you very much for your valuable and constructive comments and suggestions. Although we appreciate all the additional references you have recommended, the NHESS journal allows for only a limited (up to 20) references for brief communications (which is the format of our manuscript). Hence in the revised version of the article we have been able to follow only some of your recommendations. The full report of the European Environment Agency (EEA, 2017), which provided a background to our manuscript, contains many more references to peer-

reviewed sources supporting our analysis and conclusions. We agree and have further stressed in the revised manuscript that a sustained and constructive engagement of all interested/affected stakeholders is critically important for ensuing coherence between disaster risk reduction and climate change adaptation at all governance levels. We appreciate and some of us have contributed to the research conducted under the context of the ESPRESSO (Enhancing synergies for disaster prevention in the European Union, grant agreement 700342) Horizon 2020 research and innovation action. Our work benefited more directly from the research and analysis conducted in the context of the PLACARD (Platform for climate adaptation and risk reduction, grant agreement 653255) H2020 action.
* * *

---

## Author Comment (AC2) · 18 Aug 2018

Reviewer's comment: The authors present a brief communication related to the "coherence between CC adaptation and DRR through policies, methods and practices in Europe" based on the recently published EEA report (European Environment Agency, 2017). While the overall aim of this brief communication is not doubted, it remains a bit open to me (1) what the aim of this manuscript should be apart from advertising the original EEA report, and (2) to what extent the European research landscape is comprehensively mirrored by the authors. As such, most of the cited references

are EC and EEA reports and policy briefs, in contrast, sources other than these are widely neglected (such as e.g., OECD). As such, a brief communication in NHESS could take a wider or more focused point of view than the present one; at the moment the manuscript reads more in the way of an extended abstract of the original report. Consequently, lots of the statements made are not underpinned by references, and if they repeatedly refer to EEA reports again.

Authors' response: Thank you for your insightful comments and suggestions. Our manuscript was devised as a brief communication that summarises the process and outcomes of a comprehensive review of policies, methods and practices that characterise the intersection of disaster risk reduction (DRR) and climate change adaptation (CCA) (EEA, 2017). As for the (choice and number of) references included in our manuscript, the NHESS allows for only a limited (up to 20) references in brief communications. In the manuscript we have included references that best underpinned the our arguments and propositions. The full EEA report contains many more references to peer-reviewed sources that support our analysis and conclusions.

Reviewer's comment: The title should in my opinion focus a bit more on the content, which is not "Europe" but the Netherlands, Spain, France, the UK and Switzerland (why this selection?). As such, the manuscript shows good examples (or best practice) of CC adaptation and DRR strategies in European countries, but the focus on some countries means in turn that other examples and good practices are missing. Moreover, the examples shown in Table 1 are also a result of the respective legal regulations in the countries reported. To give an example, the multi-level risk governance reported from Switzerland is a result of the constitution of the Swiss Federation, and not as such the result of specific arrangements in DRR or CC adaptation, and cannot be similarly put to other governmental contexts such as the French one, where decisions are made by definition in a more centralized manner.

Authors' response: Our review covered all member and cooperating counties of the European Environment Agency (EEA). The process of drafting the EEA report involved

a survey among the European Environment Information and Observation Network (EIONET, www.eionet.europa.eu) countries and an expert workshop that engaged academic experts, policy advisers and policymakers closely involved in designing and implementing CCA and/or DRR policies and plans. We collected many contributions as for which examples of good practice to include in the report, and have chosen those that met our criteria. Our choices have been positively evaluated by external reviewers (including the representatives of the EIONET countries) and members of an advisory steering group that we established for this purpose. We are aware of the fact that the CCA and DRR practices are shaped by and/or reflect legislative and regulatory environments in any given country. Critical reviews like ours but also those conducted by others including OECD, EC and UNISDR serve as examples of practice that might be followed in other contexts, within the given limits.

Reviewer's comment: In the Introduction section the authors refer to initiatives other than those of EEA, such as those from EC JRC, ETC/CCA, DG Climate Action etc. On the international level, the UN Office for DRR and the Sendai Framework are particularly mentioned. Here, it would be good to also include some challenges behind such European and international actions, as recently shown by Wymann von Dach (2017) or Zimmermann and Keiler (2015) with respect to CC adaptation and DRR for European mountain regions, or OECD (2017).

Authors' response: The Sendai Framework for Disaster Risk Reduction is a commitment to a transformative change in how natural and human-made risks are dealt with (Mysiak et al., 2016; van der Vegt et al., 2015; Wahlström, 2015). SFDRR made substantial reduction of disaster losses a top priority of international efforts. To assess the progress toward this end, DRR community will have to substantially improve the practice of damage and loss assessment, including those damage and losses that may only materialize as the human-induced climate change further unfolds. For years, the United Nations Office for Disaster Risk Reduction (UNISDR) and the international community have worked to improve our knowledge of risks and to promote a culture of

evidence- and evidence-based DRR. We agree that the Sendai Framework does not solve all challenges. For example, management of increasingly interconnected risks necessitates a new policy approach that takes into account complex networks and flows of goods and services, people and capital. The new approach needs to address interconnected pathways through which 'risks can accumulate, propagate and culminate in a much greater scale of effects' (OECD, 2011). However, a critical analysis of the Framework itself is beyond the scope of our manuscript.

Reviewer's comment: Similarly, it would be nice to see some definitions here (how do the authors of the EEA study define resilience, vulnerability, and risk?).

Authors' response: The limited length of the NHESS brief communications does not make it possible to include complementary information such as definitions of basic concepts but these are explained in full depth in the EEA report that provided a background to our manuscript.

Reviewer's comment: Moreover, some of the facts given are more platitudes than proven scientific facts – at least in the way they are presented here. To give another example, the authors state that "Along with climate change, growing population and wealth, developments in hazard-prone areas, and the deteriorated status of natural ecosystems [. . .] increasing exposure and vulnerability" can be observed. This is a tricky statement, and has recently been debated (by European scholars) in the scientific literature. The manuscript would surely gain in content and rationale power if we could see some examples here – there are lots of recent works from the UK, the Netherlands or Switzerland and Austria around (such as, for example, Jongman et al. (2014) or Fuchs et al. (2015), both even published in the target journal). This would additionally help to close the still existing gap between policy advice and sciences without narrowing the aims of the original report.

Authors' response: Our statement that increased exposure and vulnerability, as well as observed climate change are responsible for upward trends in economic losses caused

by weather and climate-related disasters is correct and in line with numerous empirical studies (including those of Jongman et al 2014 and Fuchs et al. 2015) and the IPCC AR5 conclusion. Long-term trends in economic disaster losses, adjusted for wealth and population increases, have not been unambiguously attributed to climate change, although climate change amplified hazard may also have played a role.

Reviewer's comment: Similarly, section 2 seems to be a bit too narrowly focused on the EC and EEA contexts (page 3, lines 9-21). Consequently, the text is missing some concrete details or particular arguments; sentences such as the "levels of vulnerability, which include sensitivity or susceptibility to harm and lack of capacity to cope and adapt, are changing as our societies are transformed in terms of demography, wealth, cohesion and use of technology. Notwithstanding the importance of the quality-assured, systematically collected and thorough records on impacts of natural hazards, the loss data systems in Europe are fragmented and inconsistent. Empirical and evidence-based risk analysis and assessment are a vital part of CCA and DRR efforts" are a good example, also here there have been made (not only recently) some efforts from the scientific side, spanning from publications such as those of Barredo (2007, 2009, 2010) or Paprotny et al. (2018) on the European scale to national reports from different European countries, all of which are worth being noted. Alternatively, the authors should clearly state at the very beginning that the focus of this manuscript is a selection of European countries and initiatives, with an emphasis on EEA activities.

Authors' response: Given the limited length of the brief communication, not all topics that we address in our manuscript could be explained in depth. In the full report (EEA, 2017) we have dedicated an entire chapter (4.2 4.2 Disaster loss data in the European Union) to review the existing data sources and describe the practical and methodological challenges associated with collecting and analysing disaster loss data. Except for the article by Paprotny et al. (2018) which was published only after the report was released, all references that you mention are included in our review.

Reviewer's comment: Finally, section 3 reads like a document of political decision mak-
ers with (from a scientific perspective) lots of "common sense" statements, which is of course ok in a framework like the one of the original study, but it would not be amiss to be more specific in the "brief communication" to be published in a scientific journal such as NHESS.

Authors' response: In our brief communication we have outlined practical opportunities to improve the coherence between CCA and DRR, building upon the extensive evidence collected and described in full depth in the EEA (2017). Our conclusions may not be exhaustive but they are rooted in an extensive review and consultation. Our findings have been devised as a contribution to the ongoing review of the EU Climate Adaptation Strategy and the EU Action Plan on SFDRR.

Reviewer's comment: The topic itself is definitely worth being published in NHESS, and I would like to encourage the authors to undertake some revisions in the directions outlined above. As a result, I kindly would like to suggest to revise this piece of work in order to better show the relation to existing scientific works recently published (also in NHESS), and consequently to lessen the focus on EC and EEA initiatives (this can be gathered easily by the interested reader through the original version of the EEA report (European Environment Agency, 2017)).

Authors' response: Thank you, we have revised the manuscript considering yours and the other reviewer's suggestions.

References EEA, 2017. Climate change adaptation and disaster risk reduction in Europe - Enhancing coherence of the knowledge base, policies and practices, 15/2017. European Environment Agency, Copenhagen (Denmark). Mysiak, J., Surminski, S., Thieken, A., Mechler, R., Aerts, J.C.J.H., 2016. Sendai Framework for Disaster Risk Reduction – success or warning sign for Paris? Nat. Hazards Earth Syst. Sci. 16, 2189–2193. OECD, 2011. OECD Reviews of Risk Management Policies Future Global Shocks. Improving risk governance. ganisation for Economic Co-operation and Development, Paris. van der Vegt, G.S., Essens, P., Wahlström, M., George, G., 2015.

Managing risk and resilience. Acad. Manag. J. 58, 971–980. Wahlström, M., 2015. New Sendai Framework Strengthens Focus on Reducing Disaster Risk. Int. J. Disaster Risk Sci. 6, 200–201. doi:10.1007/s13753-015-0057-2

---

## Author Response (AR1)

**nhess-2018-80**

**19/10/2018**

**Brief communication: Strengthening coherence between climate change adaptation and disaster risk reduction through policies, methods and practices in Europe**

Jaroslav Mysiak, Sergio Castellari, Blaz Kurnik, Rob Swart, Patrick Pringle, Reimund Schwarze, Henk Wolters, Ad Jeuken, and Paul van der Linden

**Authors' response to the Editor(s)**

Dear Dr. Fuchs,

Thank you for your comments and suggestions. We have revised the manuscript, while paying attention to the comments of the two reviewers. Further down in this document you may find the manuscript with the revisions/modifications in track-change mode. We have

- thoroughly revised the manuscript while preserving the format (short communication),
- moved the references to regulatory and legislative documents to the supplementary material, and included instead references to peer-reviewed literature sources,
- moved parts of the text from manuscript to supplementary material and included the definitions of the main terms used.

Hereby we would like to thank the both reviewers and you for your help and insightful comments. Looking forward to hearing from you,

My best regards, Jaroslav Mysiak on behalf of all authors.

**Editor's comments to the Authors (28 Aug 2018)**

Editor Decision: Reconsider after major revisions (further review by editor and referees) (28 Aug 2018) by Sven Fuchs

Dear colleagues,

meanwhile we received two comments of referees as well as your public answers to their comments. As you may have recognized, there is a strong agreement among the referees that your short communication would gain if you include some of the issues on CCA and DRR in Europe which is beyond the statements made in the EEA report. In contrast, you argued that during the review process of the report considerable consultation with stakeholders from science and policy took place, which lead to the final statements made in the final (EEA) report. As you have stated in your response to referee #2, you have already revised the manuscript considering the suggestions made by the referees, also including the issues raised by referee #1 (stakeholder engagement…). So I kindly invite you to upload the revised version so that we can have a second round. Considering the comment of referee #2 about the definitions: you could also think about an "online supplement" or "appendix" to your article, this will not increase the page numbers, nr the limit with respect to references. I would like to thank you for your patience during the open discussion phase, and I am looking forward to receiving your finalized manuscript.

Kind regards,

Sven Fuchs (Editor NHESS)

[revised manuscript text omitted]